# Z-FOLD: A Frustratingly Easy Post-Training Quantization Scheme for LLMs

**Yongkweon Jeon,**[*,†]   **Chungman Lee,**[*]   **Kyungphil Park,**[*]   **Ho-young Kim**
Samsung Research
{dragwon.jeon, chungman.lee, k_phil.park, hoyoung4.kim}@samsung.com

## Abstract

Efficient inference has become crucial for hyper-scale AI models, including large language models, as their parameter count continues to increase for enhanced performance. This necessity holds true regardless of the computing environment, whether it be mobile devices or cloud servers. Quantization emerges as a solution to alleviate the computational burden during inference. By representing models with a reduced bit-width, quantization minimizes the frequency of DRAM access while fully exploiting the parallelism of operations through a dense matrix format. Consequently, quantized models achieve low end-to-end latency and optimize resource utilization by addressing both memory and computing bottlenecks. In this paper, we propose a straightforward post-training quantization scheme, called Z-FOLD, that fully utilizes the feature of the Transformer structure widely employed in large language models. The code will be available at https://github.com/SamsungLabs/Z-Fold.

## 1 Introduction

The Transformer (Vaswani et al., 2017) models have revolutionized machine learning across various domains by leveraging attention mechanisms (Bahdanau et al., 2014). Neural language processing (NLP) tasks, including large language models (LLMs) and neural machine translation (NMT), heavily rely on the Transformers, extending their impact to vision tasks as well (Khan et al., 2022; Rombach et al., 2022). In essence, hyperscale AI models heavily depend on the Transformer architecture.

To enhance the performance of general language tasks, large language models progressively increase the number of parameters, which are achieved with various techniques such as preventing performance saturation or enabling hyperscale model to be trained; layer dropping (Zhang

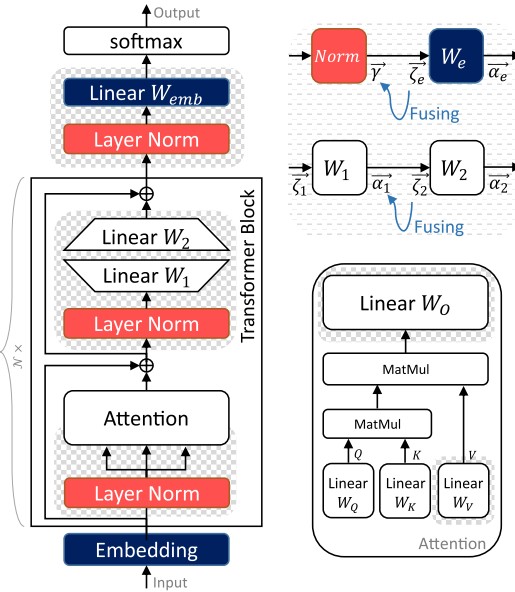

Figure 1: The Transformer Architecture (Pre-LayerNorm). We fully exploit the feature of the architecture to prevent accuracy drop from lowered bit-precision of weights.

and He, 2020), pipeline parallelism (Huang et al., 2019), model parallelism (Shoeybi et al., 2019), pre-normalization (Xiong et al., 2020), and AdamW (Loshchilov and Hutter, 2017) are employed and resulting in models with up to 100 trillion parameters.

Nonetheless, these hyperscale models encounter limitations that hinder their applicability. Among the key challenges are the memory and computational costs associated with processing an extensive number of parameters; these lead to increased end-to-end latency and power consumption, regardless of the available computing resources. Specifically, a considerable number of parameters need to be transferred between memory and processors, and computed with long input sequences. Thus, hyperscale models impose a burden on hardware during inference, despite the performance benefits of parameter scaling. Consequently, efficient

---

[*]Equal Contribution, [†]Corresponding Author

inference techniques have gained significant attention as much as for scaling out the models, whether in resource-constrained (*e.g.*, edge devices) or resource-abundant (*e.g.*, cloud server) environments.

Quantization emerges as a promising and attractive solution among various techniques aimed at improving model efficiency (*e.g.*, pruning (Frantar and Alistarh, 2023), low-rank approximation (Chen et al., 2018) and knowledge distillation (Hsieh et al., 2023)). By representing models with a low bit-width, quantization reduces the frequency of DRAM access, while fully leveraging the parallelism of operations through a dense matrix format. As a result, quantized models alleviate the bottleneck caused by both memory and computational overheads.

In this paper, we propose a post-training quantization method for LLMs, called Z-FOLD, which exploits the feature of the Transformer architecture (pre-LayerNorm) to enhance the quantized models without introducing additional quantization parameters or computational costs. In a nutshell, we quantize weights into low bit-width (down to 2-bit) using more parameters ($\zeta$) than existing approaches. This contributes to improving quantized networks by further minimizing the loss perturbation as a result of quantization. However, we fold or fuse these additional parameters ($\zeta$) into other existing parameters ($\alpha$ or $\gamma$; see Figure 1) in advance of inference, to avoid imposing further hardware costs. As a result, our approach achieves state-of-the-art performance among quantized LLMs without any additional parameters or hardware costs.

## 2 Related Works

### 2.1 Quantization

When the entire dataset is available, quantization-aware training (QAT) (Jin et al., 2021) could be useful, potentially outperforming post-training quantization (PTQ). However, given the vast number of parameters to be optimized in LLMs, retraining them with the entire dataset would not be a realistic option. QAT with few samples also suffers from overfitting due to its extensive search space. Consequently, most approaches for quantizing LLMs rely on PTQ with either a calibration set (few-shot) or without any dataset (data-free quantization). By limiting the search space to the region adjacent to the convergence point of pre-trained models, overfitting caused by few data can be prevented in PTQ.

The loss surface of pre-trained models can be approximated using the Taylor series. Assuming successful convergence and generalization of the pre-trained model, we can simplify the relationship between loss degradation ($\Delta\mathcal{L}$) and perturbation of flattened weights $\Delta\boldsymbol{w}$ (*i.e.*, $\Delta\boldsymbol{w} \triangleq \mathrm{vec}(\Delta\boldsymbol{W})$) by quantization to a linear as follows (LeCun et al., 1989; Nagel et al., 2020):

$$\Delta\mathcal{L} \approx \frac{1}{2}\Delta\boldsymbol{w}^\mathsf{T} \cdot \mathbf{H}^{(w)} \cdot \Delta\boldsymbol{w}, \qquad (1)$$

where $\mathbf{H}^{(w)}$ denotes the Hessian matrix of weights. When quantizing models in the absence of datasets, the Hessian can be approximated to $c \cdot \mathbf{I}$, where $c$ and $\mathbf{I}$ denote a constant and the Identity matrix, respectively. We thus set the *out_channel*-wise step size $\boldsymbol{\alpha} \in \mathbb{R}^{d_{out} \times 1}$ for the weights $\boldsymbol{W} \in \mathbb{R}^{d_{in} \times d_{out}}$ to minimize weight perturbation as follows:

$$\boldsymbol{\alpha}^* = \arg\min_{\boldsymbol{\alpha}} \|\boldsymbol{W} - \boldsymbol{W}_{int} \cdot \boldsymbol{A}\|_F, \qquad (2)$$

$$\boldsymbol{W}_{int} = \mathrm{clip}\left(\lfloor \boldsymbol{W} \cdot \boldsymbol{A}^{-1} \rceil, n, p\right), \qquad (3)$$

where $\lfloor\cdot\rceil$ represent the round-to-nearest function and $\boldsymbol{A}$ denotes $\boldsymbol{A} = \mathrm{diag}(\boldsymbol{\alpha})$. And $n$ and $p$ represent the lower and upper bounds, respectively, for the integer weights $\boldsymbol{W}_{int}$. Once $\boldsymbol{\alpha}^*$ is determined, we employ the nearest-rounding scheme to map the weights onto uniformly distributed quantization grids, which we refer to as *MMSE* quantization. Note that, the step size $\boldsymbol{\alpha}$ can be set simply as *min-max* quatization as follows:

$$\boldsymbol{\alpha} = \frac{\max(\boldsymbol{W}) - \min(\boldsymbol{W})}{2^{bit} - 1}. \qquad (4)$$

where $\min()$ and $\max()$ return the minimum and maximum values of each column respectively.

Based on the step size $\boldsymbol{\alpha}$ set by *MMSE* or *min-max*, we can further optimize the quantized network by various approaches approximating the Hessian with the calibration set (few-shot), which leads quantized networks to be more robust than data-free quantization (Nagel et al., 2020; Li et al., 2021; Hubara et al., 2021; Frantar and Alistarh, 2022; Jeon et al., 2023). Among them, OPTQ (Frantar et al., 2023), also known as GPTQ, achieves reasonable performance for quantized LLMs without gradient-based optimization in a relatively short period by simplifying *Optimal Brain Quantization* (Frantar and Alistarh, 2022). Consider $\boldsymbol{W}^\mathsf{T} \in \mathbb{R}^{d_{out} \times d_{in}}$ as the transposed weight matrix and $\boldsymbol{X} \in \mathbb{R}^{d_{in} \times n}$ as the input batch of the layer,

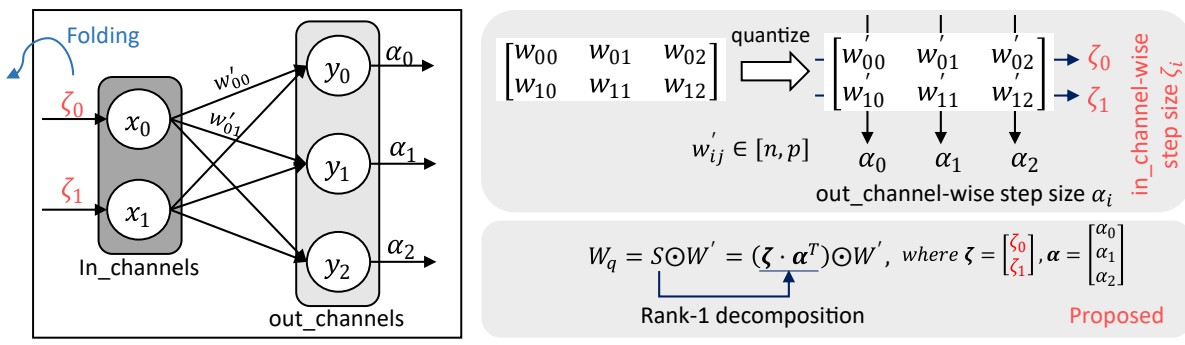

Figure 2: Decomposing the step size $\boldsymbol{S}$ into $\boldsymbol{\zeta}$ and $\boldsymbol{\alpha}$, followed by folding $\boldsymbol{\zeta}$ to the preceding layer (if available)

where $d_{in}$ and $n$ represent the input dimension and input sequence length, respectively. By assuming there is no interaction between each row of $\mathbf{W}^\mathsf{T}$, OPTQ approximates the Hessian $\mathbf{H}$ corresponding to each row of $\mathbf{W}^\mathsf{T}$ as

$$\mathbf{H} \approx 2\boldsymbol{X}\boldsymbol{X}^\mathsf{T} + \lambda\mathbf{I} \in \mathbb{R}^{d_{in} \times d_{in}}. \quad (5)$$

We thus have $\mathbf{H}^{(w)}$ as $\mathbf{H}^{(w)} = \mathbf{I} \otimes \mathbf{H}$, where $\otimes$ is the Kronecker product. OPTQ quantizes $\boldsymbol{W}^\mathsf{T}$ on column-by-column in ascending order (0 to $d_{in}$-1). Whenever each column of weights $\boldsymbol{w}_i \in \mathbb{R}^{d_{out} \times 1}$ is quantized, they measure the error $\boldsymbol{e}$ as

$$\boldsymbol{e} = (\boldsymbol{w}_i - \boldsymbol{w}_i^q)/\mathbf{H}_{ii}^{-1}, \quad (6)$$

where $\mathbf{H}^{-1} = \mathrm{Cholesky}(\mathbf{H}^{-1})^\mathsf{T}$, and then all the remaining (not yet quantized) column weights $\boldsymbol{w}_j$ ($j = i$+1 to $d_{in}$-1) are updated to reflect the perturbation of quantized weights as follows:

$$\{\boldsymbol{w}_j = \boldsymbol{w}_j - \boldsymbol{e} \cdot \mathbf{H}_{ij}^{-1}\}_{j=i+1}^{d_{in}-1}, \quad (7)$$

where $\mathbf{H}_{ij}^{-1}$ denotes the entry in the $i$th row and $j$th column of the Hessian inverse.

## 2.2 Quantization for LLMs

OPTQ enhances the robustness of quantized LLMs by approximating the Hessian matrix using a small sample of sentences, where activations are maintained in FP16 precision since they are not the bottleneck. ZeroQuant (Yao et al., 2022) adopts group-wise quantization for weights and token-wise quantization for activation to suggest a hardware-friendly quantization scheme. Despite having a larger number of scaling parameters compared to other methods, ZeroQuant adjusts the number of scaling parameters to match the hardware architecture used during inference, resulting in minimal overhead. LLM.int8() (Dettmers et al., 2022) propose a vector-wise quantization approach with

a mixed-precision scheme, in which outlier vectors are preserved as FP16. SmoothQuant (Xiao et al., 2022) and Quadapter (Park et al., 2022) have identified the challenges associated with handling the dynamic range of activations, and then suggest solutions to mitigate the variance. QuIP (Chee et al., 2023) has tried to quantize LLMs into a low bit-width (*i.e.,*, 2-bit) by introducing additional parameters (random orthogonal matrices). However, QuIP needs to store or generate such additional parameters and impose computational costs during the inference. ZeroQuant-V2 (Yao et al., 2023) has summarized and analyzed the properties of each method and bit configuration (e.g., E8A8, W4A16). While all of these approaches apply post-training quantization to LLMs, a QAT method for LLMs has been proposed in LLM-QAT (Liu et al., 2023).

## 3 Z-Fold

### 3.1 The Newly Introduced Parameter $\zeta$

To achieve even greater reductions in the loss perturbation due to quantization (Eq. (1)), we introduce new scaling factor $\boldsymbol{\zeta} \in \mathbb{R}^{m \times 1}$ corresponding the *in_channel* in addition to existing *out_channel*-wise scaling factors $\boldsymbol{\alpha} \in \mathbb{R}^{n \times 1}$. The scaling factor $\boldsymbol{\zeta}$ is then fused or folded into $\boldsymbol{\alpha}$ of the previous layer in advance of inference to eliminate the overhead by additional parameters $\boldsymbol{\zeta}$, which we call Z-Fold[1] (see Figure 2). In essence, our objective is to obtain the bi-directional (*in* and *out*) step size matrix $\boldsymbol{S}$, which can be formulated as follows:

$$\boldsymbol{S}^* = \arg\min_{\boldsymbol{S}} \Delta\boldsymbol{w}^\mathsf{T} \cdot \mathbf{H}^{(w)} \cdot \Delta\boldsymbol{w}, \quad (8)$$

$$\Delta\boldsymbol{w} = \mathrm{vec}(\boldsymbol{W} - \boldsymbol{S} \odot \boldsymbol{W}_{int}) \quad (9)$$

$$\boldsymbol{W}_{int} = \mathrm{clip}\left(\lfloor \boldsymbol{W} \oslash \boldsymbol{S} \rceil, n, p\right), \quad (10)$$

where $\oslash$ and $\odot$ denote the Hadamard division and product, respectively. To determine $\boldsymbol{S}^*$ in Eq. (8),

---

[1]step size-Folding

**Algorithm 1** Alternating Least Square for $\boldsymbol{S}$

**Input**: $\boldsymbol{W} \in \mathbb{R}^{d_{in} \times d_{out}}$

**Output**: $\boldsymbol{\alpha} \in \mathbb{R}^{d_{out} \times 1}, \boldsymbol{\zeta} \in \mathbb{R}^{d_{in} \times 1}, (\boldsymbol{\zeta} \cdot \boldsymbol{\alpha}^{\intercal} = \boldsymbol{S})$

1: **procedure** Z-FOLD($\boldsymbol{W}, \mathbf{H} = 2\boldsymbol{X}\boldsymbol{X}^{\intercal} + \lambda\mathbf{I}$)
2:     Initialize $\boldsymbol{\alpha}$ and $\boldsymbol{\zeta}$
3:     **repeat**
4:         $\boldsymbol{W}_{int}$=Quantize($\boldsymbol{W}, \boldsymbol{\zeta} \cdot \boldsymbol{\alpha}^{\intercal}$)
5:         $\boldsymbol{\zeta}$ = Least_Squares($\boldsymbol{W}^{\intercal}, (\boldsymbol{W}_{int} \cdot \mathrm{diag}(\boldsymbol{\alpha}))^{\intercal}, \mathbf{I}$)
6:         $\boldsymbol{W}_{int}$=Quantize($\boldsymbol{W}, \boldsymbol{\zeta} \cdot \boldsymbol{\alpha}^{\intercal}$)
7:         $\boldsymbol{\alpha}$ = Least_Squares($\boldsymbol{W}, \mathrm{diag}(\boldsymbol{\zeta}) \cdot \boldsymbol{W}_{int}, \mathbf{H}$)
8:     **until** converged
9:     return $\boldsymbol{\alpha}, \boldsymbol{\zeta}$
10: **procedure** QUANTIZE($\boldsymbol{W},\boldsymbol{S}$)
11:     return clip $(\lfloor \boldsymbol{W} \oslash \boldsymbol{S} \rceil, n, p)$
12: **procedure** LEAST_SQUARES($\boldsymbol{W}, \tilde{W}, \mathbf{H}$)
13:     **for** each column vector $(\boldsymbol{w}_i, \tilde{\boldsymbol{w}}_i)$ in $(\boldsymbol{W}, \tilde{\boldsymbol{W}})$ **do**
14:         $y_i = (\tilde{\boldsymbol{w}}_i^{\intercal}\mathbf{H}\tilde{\boldsymbol{w}}_i)^{-1}\tilde{\boldsymbol{w}}_i^{\intercal}\mathbf{H} \cdot \boldsymbol{w}_i$
15:     return $\boldsymbol{y}$

we decompose the matrix $\boldsymbol{S}$ into two vectors (the rank-1 approximation) by alternating least square (ALS) factorization (Zachariah et al., 2012). Specifically, we determine those value of $\boldsymbol{\alpha} \in \mathbb{R}^{n \times 1}$ and $\boldsymbol{\zeta} \in \mathbb{R}^{m \times 1}$ as follows:

$$\Delta\boldsymbol{W} = \|\boldsymbol{W} - \boldsymbol{Z} \cdot \boldsymbol{W}_{int} \cdot \boldsymbol{A}\|_2^2, \quad (11)$$

$$\text{when } \boldsymbol{Z} \triangleq \mathrm{diag}(\boldsymbol{\zeta}) \text{ and } \boldsymbol{A} \triangleq \mathrm{diag}(\boldsymbol{\alpha}), \quad (12)$$

which can be achieved by vector-wise least squares. Considering a vector $\boldsymbol{w}_i \in \mathbb{R}^{m \times 1}$ representing the $i$th column vector of $\boldsymbol{W} \in \mathbb{R}^{m \times n}$, we can leverage the weighted least squares to determine $\boldsymbol{\alpha}$ using the Hessian as follows:

$$\alpha_i = (\tilde{\boldsymbol{w}}_i^{\intercal}\mathbf{H}\tilde{\boldsymbol{w}}_i)^{-1}\tilde{\boldsymbol{w}}_i^{\intercal}\mathbf{H} \cdot \boldsymbol{w}_i \quad (13)$$

where $\tilde{\boldsymbol{w}}_i$ denotes $\tilde{\boldsymbol{w}}_i = \mathrm{diag}(\boldsymbol{\zeta}) \cdot \boldsymbol{w}_{int,i}$. The weighted least square is used only for $\boldsymbol{\alpha}$ and alternately refined with $\boldsymbol{\zeta}$. Note that, because the weights sharing $\boldsymbol{\zeta}$ also share the entries of the Hessian, whether or not the Hessian is used to determine $\boldsymbol{\zeta}$ does not affect the result for $\boldsymbol{\zeta}$.

Algorithm 1 provides a detailed overview of the factorization algorithm. We initialize $\boldsymbol{\zeta}$ as all-1 vector and then employ *Min-Max* quantization for $\boldsymbol{\alpha}$ as in Eq. (4) (*Line 2*). Subsequently, $\boldsymbol{\alpha}$ and $\boldsymbol{\zeta}$ are alternately updated using a least square solution until convergence (*Line 3–8*), such that minimizing the loss as in Eq. (8). Note that column-wise least square (*Line 12–14*) offers loop-level parallelism allowing for parallel processing via batch matrix-matrix multiplication (*i.e.,* bmm). With the

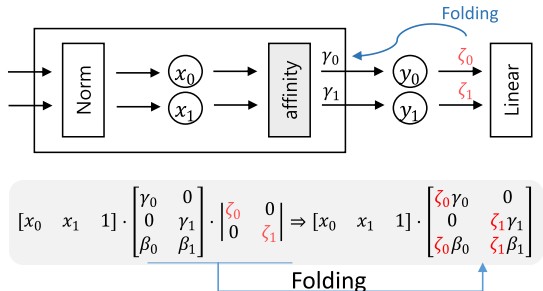

Figure 3: Norm-to-Linear Folding. The *in_channel*-wise parameters can be fused into the affine transformation parameters of the previous normalization layer.

factorized the step size from weights matrix $\boldsymbol{W}$, we obtain quantized weights matrix $\boldsymbol{W}^q$ as follows:

$$\boldsymbol{W}^q = \boldsymbol{S} \odot \boldsymbol{W}_{int} \quad (14)$$
$$= \mathrm{diag}(\boldsymbol{\zeta}) \cdot \boldsymbol{W}_{int} \cdot \mathrm{diag}(\boldsymbol{\alpha}) \quad (15)$$
$$= \boldsymbol{Z} \cdot \boldsymbol{W}_{int} \cdot \boldsymbol{A} \quad (16)$$

Our scheme also supports per-*out_channel* asymmetric quantization, in which each column weight $\boldsymbol{w}_i$ can be quantized as follows:

$$\boldsymbol{w}_i^q = \alpha(\boldsymbol{w}_{i,\text{int}} - \boldsymbol{o}) \odot \boldsymbol{\zeta} \quad (17)$$
$$\boldsymbol{w}_{i,\text{int}} = \mathrm{clip}\left(\left\lfloor \frac{\boldsymbol{w}_i \oslash \boldsymbol{\zeta}}{\alpha} \right\rceil + \boldsymbol{o}, n, p\right). \quad (18)$$

where $\boldsymbol{o}$ indicates the zero point vector (all-$o$ vector), computed by $o = -\left\lfloor \frac{\min(\boldsymbol{w}_i \oslash \boldsymbol{\zeta})}{\alpha} \right\rceil$. For asymmetric quantization, we can employ a grid search to identify the optimal values of $\boldsymbol{\alpha}$ and $\boldsymbol{o}$, aiming to minimize the loss perturbation caused by quantization. This grid search replaces the use of least squares in *Line 7* of Algorithm 1.

### 3.2 Parameter Integration ($\boldsymbol{\zeta}$-Folding)

Figures 2, 3 and 5 illustrate the mechanism of Z-FOLD. The figures demonstrate how *in_channel*-wise parameters $\boldsymbol{\zeta}$ can be integrated into either the affine transformation parameters $\boldsymbol{\gamma}$ and $\boldsymbol{\beta}$ (Norm-to-Linear) or the *out_channel*-wise parameters $\boldsymbol{\alpha}$ (Linear-to-Linear) of the preceding layer.

**Norm-to-Linear** Let us consider a latent token $\boldsymbol{x} \in \mathbb{R}^{m \times 1}$ as the input of a normalization layer followed by a linear layer, the normalized vector, denoted by $\hat{\boldsymbol{x}}$ is computed as follows:

$$\boldsymbol{y} = \boldsymbol{\gamma} \odot \hat{\boldsymbol{x}} + \boldsymbol{\beta} \quad (19)$$

where $\boldsymbol{\gamma} \in \mathbb{R}^{m \times 1}$ and $\boldsymbol{\beta} \in \mathbb{R}^{m \times 1}$ denote affine transform parameters of $\hat{\boldsymbol{x}}$ (*i.e.,* the scale and shift).

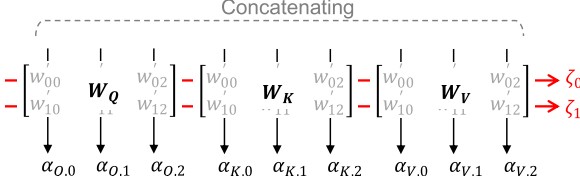

Figure 4: Concatenating $W_Q$, $W_K$, and $W_V$ to extract a mutual $\zeta$ for fusion into the preceding normalization layer.

Eq. (19) can be expressed as a matrix operation, as follows:

$$y^{\intercal} = \begin{bmatrix} \hat{x}^{\intercal} & 1 \end{bmatrix} \cdot \Gamma, \qquad (20)$$

$$\text{where } \Gamma = \begin{bmatrix} \text{diag}(\gamma) \\ \beta^{\intercal} \end{bmatrix} \triangleq \begin{bmatrix} \gamma_0 & 0 & \cdots & 0 \\ 0 & \gamma_1 & \cdots & 0 \\ \vdots & \vdots & \ddots & \vdots \\ 0 & 0 & \cdots & \gamma_{m-1} \\ \beta_0 & \beta_1 & \cdots & \beta_{m-1} \end{bmatrix}. \qquad (21)$$

The output $y$ from the normalization layer is then multiplied by quantized weights $W^q \in \mathbb{R}^{m \times n}$ of the subsequent linear layer as follows:

$$y^{\intercal} \cdot W^q = y^{\intercal} \cdot (Z \cdot W_{int} \cdot A) \qquad (22)$$

$$= \left( \begin{bmatrix} \hat{x}^{\intercal} & 1 \end{bmatrix} \cdot \Gamma \right) \cdot (Z \cdot W_{int} \cdot A) \qquad (23)$$

$$= \left( \begin{bmatrix} \hat{x}^{\intercal} & 1 \end{bmatrix} \cdot (\Gamma \cdot Z) \right) \cdot (W_{int} \cdot A). \qquad (24)$$

As in Eq. (24), we can update the affine transform parameters using the $\zeta$ of the subsequent linear. As a result, We obtain the revised parameters as

$$\Gamma' = \Gamma \cdot Z = \begin{bmatrix} \gamma_0 \zeta_0 & 0 & \cdots & 0 \\ 0 & \gamma_1 \zeta_1 & \cdots & 0 \\ \vdots & \vdots & \ddots & \vdots \\ 0 & 0 & \cdots & \gamma_{m-1} \zeta_{m-1} \\ \beta_0 \zeta_0 & \beta_1 \zeta_1 & \cdots & \beta_{m-1} \zeta_{m-1} \end{bmatrix}. \qquad (25)$$

Essentially, we can update the affinity parameters without relying on the dataset to compensate for the errors caused by quantization. By folding $\zeta$ into $\gamma$ and $\beta$ of the preceding normalization layer, we can reduce the loss by quantization weights without increasing the number of parameters and computational cost.

Notably, by concatenating the weight matrices of $Q$, $K$, and $V$, the shared $\zeta$ can be extracted; consequently, the shared $\zeta$ can be seamlessly incorporated into the $\gamma$ and $\beta$ of the preceding normalization layer (see Figure 4).

**Linear-to-Linear** It can be easily demonstrated that the *in_channel*-wise step size $\zeta$ can be fused into the *out_channel*-wise step size $\alpha$ of the preceding layer, provided there is no non-linear function

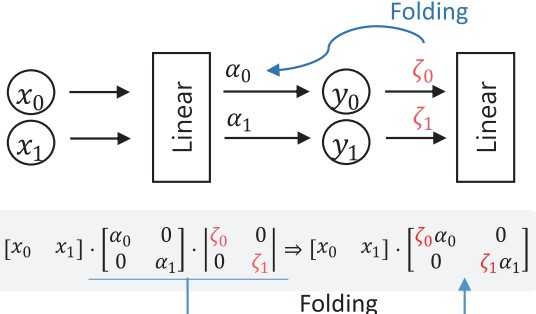

Figure 5: Linear-to-Linear Folding. The quantization parameters between two adjacent linear layers can be fused if there is no non-linear function except ReLU.

---

**Algorithm 2** Fine Tuning Each Layer

---

**Input**: $W \in \mathbb{R}^{d_{in} \times d_{out}}$, $X \in \mathbb{R}^{d_{in} \times n}$ ($Y = X^{\intercal} \cdot W$)
**Output**: $\alpha, \zeta, W^q$
 1: **procedure** FINE-TUNING WITH OPTQ($W,X$)
 2:   $H = 2XX^{\intercal} + \lambda I$
 3:   $\alpha, \zeta$=Z-FOLD($W, H$) ▷ Algorithm 1
 4:   $H^{-1} = \text{Cholesky}(H^{-1})^{\intercal}$
 5:   **for** each row vector $w_i$ in $W$ **do**
 6:     $s = (\zeta_i \odot \alpha) \in \mathbb{R}^{d_{out} \times 1}$ ▷ the step size of $w_i$
 7:     $w_i^q = s^{\intercal} \cdot \text{clip} \left( \lfloor w_i \oslash s^{\intercal} \rceil, n, p \right)$
 8:     $e = (w_i - w_i^q)/H_{ii}^{-1}$
 9:     $\{ w_j = w_j - e \cdot H_{ij}^{-1} \}_{j=i+1}^{d_{in}-1}$
10:   **return** $\alpha, \zeta, W^q$

---

excepting ReLU between adjacent two linear layers. Considering $W_1^q \in \mathbb{R}^{n \times m}$ and $W_2^q \in \mathbb{R}^{m \times n}$ as the weights of two consecutive linear layers and $x \in \mathbb{R}^{n \times 1}$ as the input of the first layer, the output can be computed as follows:

$$(x^{\intercal}) \cdot (W_1^q A_1) \cdot (Z_2 W_2^q A_2) \qquad (26)$$

$$= (x^{\intercal}) \cdot (W_1^q A_1 Z_2) \cdot (W_2^q A_2) \qquad (27)$$

As shown in Figure 1, it can also fuse the pair of layers $W_O$ and $W_V$ in addition to $W_1$ and $W_2$. Suppose $Y$ and $X$ denote $Y = \text{softmax}(Q, K^{\intercal})$ and the input of the attention layer, respectively. Then, the output of linear, $W_O$, remained consistent even after parameter incorporation as follows:

$$(Y^{\intercal}) \cdot (X^{\intercal} \cdot W_V^q A_V) \cdot (Z_O W_O^q A_O) \qquad (28)$$

$$= (Y^{\intercal}) \cdot (X^{\intercal} \cdot W_V^q A_V Z_O) \cdot (W_O^q A_O) \qquad (29)$$

### 3.3 Fine-Tuning with a Calibration Set

After determining the bi-directional step size matrix $S$ as in Algorithm 1, we further optimize quantized LLMs with a small calibration set by performing optimization using the approximated Hessian (Nagel et al., 2020; Li et al., 2021; Hubara

Table 1: Quantization performance of Z-FOLD on OPT models (Perplexity ↓).

(a) WikiText-2

| Method | Precision | 125M | 350M | 1.3B | 2.7B | 6.7B | 13B | 30B |
|---|---|---|---|---|---|---|---|---|
| Baseline | FP16 | 27.65 | 22.00 | 14.63 | 12.47 | 10.86 | 10.13 | 9.56 |
| RTN[†] | | 37.28 | 25.94 | 48.17 | 16.92 | 12.10 | 11.32 | 10.98 |
| OPTQ[†] | INT4 | 31.12 | 24.24 | 15.47 | 12.87 | 11.39 | 10.31 | 9.63 |
| **Z-FOLD** | | **30.18** | **23.64** | **14.92** | **12.41** | **10.95** | **10.15** | **9.52** |
| RTN[†] | | 1.3e3 | 64.57 | 1.3e4 | 1.6e4 | 5.8e3 | 3.4e3 | 1.6e3 |
| OPTQ[†] | INT3 | 53.85 | 33.79 | 20.97 | 16.88 | 14.86 | 11.61 | 10.27 |
| **Z-FOLD** | | **37.19** | **28.18** | **16.27** | **13.53** | **11.43** | **10.65** | **9.89** |
| RTN | | 5.5e3 | 2.8e4 | 1.1e5 | 9.5e3 | 2.8e4 | 1.9e5 | 1.7e5 |
| OPTQ | INT2 | 5.6e3 | 1.7e4 | 8.0e3 | 9.0e3 | 3.4e3 | 348.2 | 61.53 |
| **Z-FOLD** | | **152.3** | **127.9** | **36.91** | **28.00** | **19.36** | **15.84** | **13.13** |

(b) PTB

| Method | Precision | 125M | 350M | 1.3B | 2.7B | 6.7B | 13B | 30B |
|---|---|---|---|---|---|---|---|---|
| Baseline | FP16 | 38.99 | 31.08 | 20.29 | 17.97 | 15.77 | 14.52 | 14.04 |
| RTN[†] | | 53.89 | 36.79 | 57.30 | 31.05 | 18.84 | 16.51 | 15.40 |
| OPTQ[†] | INT4 | 45.17 | 34.52 | 21.85 | 19.14 | 16.56 | 14.94 | 14.26 |
| **Z-FOLD** | | **42.83** | **34.03** | **21.11** | **18.45** | **15.85** | **14.67** | **14.21** |
| RTN[†] | | 1.4e3 | 88.04 | 1.3e4 | 1.4e4 | 5.7e3 | 2.8e3 | 1.2e3 |
| OPTQ[†] | INT3 | 73.19 | 47.08 | 32.10 | 24.81 | 21.88 | 16.68 | 15.36 |
| **Z-FOLD** | | **53.74** | **40.23** | **23.15** | **20.30** | **16.67** | **15.19** | **14.58** |
| RTN | | 4.3e3 | 2.8e4 | 1.1e4 | 6.8e3 | 1.8e4 | 1.2e5 | 1.7e5 |
| OPTQ | INT2 | 3.8e3 | 1.5e4 | 7.1e3 | 8.6e3 | 2.4e4 | 275.7 | 87.54 |
| **Z-FOLD** | | **210.1** | **190.2** | **60.42** | **48.53** | **27.89** | **23.57** | **19.15** |

(c) C4

| Method | Precision | 125M | 350M | 1.3B | 2.7B | 6.7B | 13B | 30B |
|---|---|---|---|---|---|---|---|---|
| Baseline | FP16 | 26.56 | 22.59 | 16.07 | 14.34 | 12.71 | 12.06 | 11.44 |
| RTN[†] | | 33.91 | 26.21 | 24.51 | 18.43 | 14.36 | 13.36 | 13.46 |
| OPTQ[†] | INT4 | 29.22 | 24.63 | 16.97 | 15.00 | 13.18 | 12.26 | 11.57 |
| **Z-FOLD** | | **27.96** | **24.12** | **16.50** | **14.65** | **12.86** | **12.15** | **11.50** |
| RTN[†] | | 834.0 | 55.49 | 5.2e3 | 1.1e4 | 5.3e3 | 3.1e3 | 1.4e3 |
| OPTQ[†] | INT3 | 42.41 | 31.33 | 21.63 | 18.17 | 17.14 | 13.34 | 12.23 |
| **Z-FOLD** | | **32.75** | **27.33** | **17.49** | **15.64** | **13.36** | **12.47** | **11.74** |
| RTN | | 3.7e3 | 1.6e4 | 7.7e3 | 7.7e3 | 1.4e4 | 9.7e4 | 5.6e4 |
| OPTQ | INT2 | 2.3e3 | 5.5e3 | 4.3e3 | 4.2e3 | 568.1 | 124.5 | 29.03 |
| **Z-FOLD** | | **106.8** | **87.96** | **34.53** | **26.73** | **21.60** | **17.11** | **14.10** |

[†] Results are taken from (Frantar et al., 2023).
[*] In OPT-350M, post-LayerNorm architecture has been used so that we only apply Z-FOLD for $W_O$ and $W_2$ (see Figure 1).

et al., 2021; Wei et al., 2022; Frantar et al., 2023; Jeon et al., 2023). Among them, we utilize OPTQ, which is a time-efficient solution for hyper-scale models such as LLMs. In essence, we utilize Z-FOLD as the initial quantization state and subsequently apply a few-shot quantization method to refine quantized models further. Algorithm 2 explains our approach when given a calibration set. We initialize quantized models by Z-FOLD, where we use the Hessian for $\alpha$ to minimize the degrada-

Table 2: Quantization performance of Z-FOLD on LLaMA models (Perplexity ↓).

| Method | Precision | WikiText-2 | | | PTB | | | C4 | | |
|--------|-----------|------|------|------|------|------|------|------|------|------|
| | | 7B | 13B | 30B | 7B | 13B | 30B | 7B | 13B | 30B |
| Baseline | FP16 | 5.68 | 5.09 | 4.10 | 8.80 | 8.07 | 7.30 | 7.08 | 6.61 | 5.98 |
| RTN | | 6.29 | 5.53 | 4.54 | 9.70 | 8.63 | 7.70 | 7.73 | 6.99 | 6.33 |
| OPTQ | INT4 | 6.11 | 5.35 | 4.46 | **9.31** | 8.42 | 7.53 | **7.43** | 6.84 | 6.20 |
| **Z-FOLD** | | **6.07** | **5.28** | **4.34** | 9.35 | **8.31** | **7.52** | 7.47 | **6.77** | **6.13** |
| RTN | | 25.62 | 11.78 | 14.87 | 61.71 | 22.38 | 30.68 | 28.24 | 13.24 | 28.57 |
| OPTQ | INT3 | 8.11 | 6.71 | 5.68 | 12.06 | 9.99 | 8.90 | 9.56 | 8.22 | 7.36 |
| **Z-FOLD** | | **6.79** | **5.76** | **4.92** | **10.43** | **8.96** | **7.97** | **8.22** | **7.21** | **6.59** |
| RTN | | 1.1e5 | 5.7e4 | 2.7e4 | 1.2e5 | 7.8e4 | 2.6e4 | 1.1e5 | 5.9e4 | 2.8e4 |
| OPTQ | INT2 | 1.0e4 | 7703. | 2065. | 7620. | 1.4e4 | 5666. | 1031. | 1185. | 349.0 |
| **Z-FOLD** | | **14.58** | **13.25** | **9.65** | **24.36** | **20.98** | **17.01** | **15.19** | **13.10** | **10.82** |

tion of the loss (Eq. (1)) (*Line 2–3*). We then apply the OPTQ to minimize the loss further (*Line 4–9*).

Our scheme can be applied to not only uniform quantization but also multi-level binary quantization (Xu et al., 2018; Jeon et al., 2022). Both formats can be accelerated by dedicated kernels (Jeon et al., 2020; Frantar et al., 2023) when performing LLMs inference without quantizing activations. As well, Z-FOLD can be utilized in QLoRA (Dettmers et al., 2023) when quantizing and freezing the pre-trained weights of LLMs.

## 4 Experiments

### 4.1 Experimental Setup

To evaluate the performance of the proposed method, we quantize publicly available LLMs (e.g., OPT (Zhang et al., 2022), LLaMA (Touvron et al., 2023), and BLOOM (Scao et al., 2022)) using Z-FOLD. As a calibration dataset to estimate the Hessian, we use 128 random 2048 token segments from the C4 dataset (Raffel et al., 2019) as in OPTQ (Frantar et al., 2023). We conduct all experiments using a single NVIDIA A100 GPU (80 GB) and implement Z-FOLD using PyTorch.

In our experiments, we quantize only weights and retain activations as full-precision (FP16) since activations do not pose a significant bottleneck during the inference of LLMs (Frantar et al., 2023). We break the alternating updates of $\alpha$ and $\zeta$ in Z-FOLD (see lines 4-7 in Algorithm 1) when the loss perturbation $\Delta \mathcal{L}$ (see equation 1) increases after the update or the total number of updates exceeds 30 iterations.

After the quantization, we evaluate the performance of the quantized LLMs using three bench-

mark datasets (WikiText-2 (Merity et al., 2016), C4 (Raffel et al., 2019), and PTB (Marcus et al., 1993)) and several zero-shot tasks.

### 4.2 Comparison with Others

In Table 1, we summarize quantization results on OPT models of various sizes (125M to 30B). For comparison, we also summarize the results for conventional RTN and OPTQ schemes. Overall, Z-FOLD and OPTQ outperform RTN by a large margin since they minimize the loss degradation $\Delta \mathcal{L}$, rather than the weight quantization error $\Delta w$, by exploiting the estimated Hessian. Furthermore, we can observe that the proposed Z-FOLD outperforms OPTQ, and the performance gap is getting larger when quantizing models into a lower bit width.

In the case of 2-bit quantization, for example, OPTQ collapses completely (perplexity is larger than 2,000 regardless of the model size and the test dataset) while Z-FOLD still shows reasonable perplexity. The outstanding performance of Z-FOLD results from dividing quantization levels more precisely using an additional parameter $\zeta$, yet imposing no extra cost via folding (see Section 3.2). Such an outperformance can be also observed for LLaMA (see Table 2) and BLOOM models (see Table 5 in Appendix A.1).[2]

Z-FOLD performs better than OPTQ in almost all cases, and the performance gap is significant in a low bit width. We note that the quantized models obtained by OPTQ perform much worse than the full-precision model even for large-scale

---

[2]For BLOOM, GeLU is used as an activation function between the adjacent linear layers ($W_1$ and $W_2$ in Figure 1). In this case, $\zeta$ of $W_2$ cannot be folded into $W_1$ so we do not apply Z-FOLD for $W_2$.

Table 3: Quantization (INT2) results for Z-FOLD combined with AdaRound/BRECQ (Perplexity ↓)

| Method | WikiText-2 | PTB | C4 |
|---|---|---|---|
| Baseline (OPT-125M) | 27.65 | 38.99 | 26.56 |
| AdaRound | 570.5 | 877.6 | 298.6 |
| AdaRound + **Z-FOLD** | **158.8** | **227.6** | **100.6** |
| BRECQ | 53.95 | 82.19 | 41.50 |
| BRECQ + **Z-FOLD** | **45.43** | **60.00** | **36.73** |

Table 4: Ablation study for Z-FOLD on 2-bit quantization of OPT-125M (Perplexity ↓)

| Method | Quant. Grid | WikiText-2 | PTB | C4 |
|---|---|---|---|---|
| OPTQ | Min-Max | 5606. | 3829. | 2333. |
| **Z-FOLD** | | **3254.** | **2940.** | **1181.** |
| OPTQ | MMSE | 978.3 | 1192. | 628.4 |
| **Z-FOLD** | | **493.3** | **680.9** | **314.9** |
| OPTQ | Hessian | 231.1 | 375.6 | 186.0 |
| **Z-FOLD** | | **152.3** | **210.1** | **106.8** |

models (e.g., LLaMA-30B) that might be overparameterized and thus expected to be quantized with marginal performance. For example, the perplexity of the 2-bit LLaMA-30B model is 2065 while that of the full-precision model is only 4.10. Whereas, the perplexity obtained by the proposed Z-FOLD is 9.65, which means that the model quantized by Z-FOLD performs very close to the full-precision model.

### 4.3 Generalizability of Z-FOLD

We investigate the generalizability of LLMs quantized by Z-FOLD using zero-shot tasks such as LAMBADA (Paperno et al., 2016), PIQA (Bisk et al., 2020), StoryCloze (Mostafazadeh et al., 2017), and ARC (Easy and Challenge) (Clark et al., 2018). We recall that the calibration data we used for the quantization (i.e., random token segments from the C4 dataset) is excerpts from randomly crawled websites, which means that our calibration data is not task-specific (generic) and zero-shot setting is kept in our experiments.

In Tables 6 to 8 (see Appendix A.2), we summarize the zero-shot performance of quantized LLMs. From the results, we observe that the models quantized by the proposed Z-FOLD are much more robust than those obtained by conventional RTN and OPTQ approaches. In particular, for the LAMBADA task, while RTN and OPTQ completely fail the 2-bit quantization even for 30B models, Z-FOLD shows reasonable performance owing to more sophisticated quantization capability induced by $\zeta$.

### 4.4 Portability of Z-FOLD

So far, we have used Z-FOLD together with OPTQ to quantize LLMs. To show the versatility of the proposed Z-FOLD, we perform the quantization by combining Z-FOLD with other well-known quantization approaches such as AdaRound (Nagel et al., 2020) and BRECQ (Li et al., 2021) (see Table 3).

We only consider OPT-125M since AdaRound and BRECQ require more computing resources and time to complete the quantization of large-scale models than OPTQ. As in Table 3, we observe that Z-FOLD greatly enhances the quantization performance of AdaRound and BRECQ. In particular, when combined with BRECQ, the quantized model obtained by Z-FOLD shows comparable performance with the full-precision model.

### 4.5 Ablation Study

We recall that two key features of the proposed Z-FOLD are 1) additional (foldable) parameters $\zeta$ and 2) Hessian-based computation of quantization grids (i.e., $\zeta$ and $\alpha$). Note that Z-FOLD utilizes the Hessian information when determining the step size while existing works such as AdaRound and OPTQ use *MMSE* or *Min-Max* for the step size.

To examine the impact of each feature, we conduct ablation studies (see Table 4). From the results, we observe that the quantization performance can be enhanced by exploiting $\zeta$, which demonstrates the effectiveness of the proposed $\zeta$ in refining quantization grids. We also note that the performance of Z-FOLD can be improved dramatically by utilizing the Hessian **H** in computing $\zeta$ and $\alpha$. In other words, we can further optimize the quantized networks by using the Hessian **H** in searching the quantization grid (i.e., $\alpha$ and $\zeta$). Which contributes to reducing the loss perturbation $\Delta\mathcal{L}$ rather than the quantization error $\Delta W$.

## 5 Conclusion

In this paper, we have proposed a post-training quantization scheme called Z-FOLD. By capturing the unique features of the Transformer and fully leveraging its architecture (pre-LayerNorm and successivity of linear layers), we developed a simple yet powerful approach that enhances the perfor-

mance of quantized models. Our scheme stands out by quantizing models using a greater number of parameters than existing approaches, resulting in enhanced sophistication. Importantly, these additional parameters are intelligently fused into existing ones, eliminating the need for extra hardware costs. Our findings demonstrate that the Z-FOLD scheme achieves state-of-the-art performance in quantized LLMs.

# 6 Limitations

Our proposed scheme has a dependence on the target architecture although the pre-LN Transformer we target (*e.g.,* OPT, BLOOM, and LLaMA) is broadly used in generative large language models.

# Acknowledgements

We would like to thank Junhan Kim, Ph.D for his helpful discussion.

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

# A  Additional Experimental Results

In the Appendix, we provide experimental results excluded in the main text due to the page limitation.

## A.1  Results for BLOOM Models

Table 5: Quantization performance of Z-FOLD on BLOOM models (Perplexity ↓).

(a) WikiText-2

| Method | Precision | 560M | 1.1B | 1.7B | 3B | 7.1B |
|---|---|---|---|---|---|---|
| Baseline | FP16 | 22.42 | 17.69 | 15.39 | 13.48 | 11.37 |
| RTN[†] | | 25.90 | 22.00 | 16.97 | 14.76 | 12.10 |
| OPTQ[†] | INT4 | 24.03 | 19.05 | 16.48 | 14.20 | 11.73 |
| **Z-FOLD** | | **23.58** | **18.61** | **15.99** | **13.90** | **11.59** |
| RTN[†] | | 57.08 | 50.19 | 63.59 | 39.36 | 17.38 |
| OPTQ[†] | INT3 | 32.31 | 25.08 | 21.11 | 17.40 | 13.47 |
| **Z-FOLD** | | **27.16** | **21.24** | **17.75** | **15.21** | **12.35** |
| RTN | | 7.9e5 | 9.9e5 | 3.5e5 | 1.4e5 | 2.1e5 |
| OPTQ | INT2 | 2646. | 2220. | 1791. | 889.5 | 218.0 |
| **Z-FOLD** | | **76.63** | **51.41** | **33.66** | **27.69** | **19.69** |

(b) PTB

| Method | Precision | 560M | 1.1B | 1.7B | 3B | 7.1B |
|---|---|---|---|---|---|---|
| Baseline | FP16 | 43.69 | 57.96 | 30.00 | 25.34 | 20.83 |
| RTN[†] | | 51.10 | 66.85 | 33.58 | 27.68 | 22.42 |
| OPTQ[†] | INT4 | 46.97 | 62.47 | 31.84 | 26.49 | 21.67 |
| **Z-FOLD** | | **43.40** | **49.87** | **29.19** | **24.08** | **19.87** |
| RTN[†] | | 126.0 | 185.0 | 106.0 | 66.78 | 35.04 |
| OPTQ[†] | INT3 | 70.35 | 87.04 | 46.11 | 34.02 | 26.14 |
| **Z-FOLD** | | **51.43** | **58.14** | **34.44** | **26.58** | **21.29** |
| RTN | | 1.1e6 | 2.0e6 | 3.2e5 | 1.2e5 | 1.3e5 |
| OPTQ | INT2 | 4548. | 5202. | 4254. | 2619. | 579.2 |
| **Z-FOLD** | | **184.6** | **154.5** | **68.05** | **55.57** | **38.23** |

(c) C4

| Method | Precision | 560M | 1.1B | 1.7B | 3B | 7.1B |
|---|---|---|---|---|---|---|
| Baseline | FP16 | 26.60 | 22.05 | 19.49 | 17.49 | 15.20 |
| RTN[†] | | 29.89 | 24.44 | 21.26 | 18.76 | 16.06 |
| OPTQ[†] | INT4 | 28.00 | 23.25 | 20.55 | 18.10 | 15.60 |
| **Z-FOLD** | | **25.24** | **20.96** | **18.53** | **16.50** | **14.36** |
| RTN[†] | | 67.49 | 60.71 | 113.0 | 80.49 | 22.59 |
| OPTQ[†] | INT3 | 35.78 | 28.83 | 25.34 | 21.25 | 17.67 |
| **Z-FOLD** | | **27.96** | **22.77** | **19.93** | **17.51** | **15.01** |
| RTN | | 1.1e6 | 1.9e6 | 2.6e5 | 8.8e4 | 1.3e6 |
| OPTQ | INT2 | 763.9 | 699.6 | 499.4 | 332.5 | 108.0 |
| **Z-FOLD** | | **62.17** | **41.46** | **33.24** | **27.12** | **21.23** |

[†] Results are taken from (Frantar et al., 2023).

## A.2 Results on Zero-Shot Tasks

Table 6: Zero-shot performance of quantized OPT models (Accuracy ↑).

(a) LAMBADA

| Method | Precision | 125M | 350M | 1.3B | 2.7B | 6.7B | 13B | 30B |
|---|---|---|---|---|---|---|---|---|
| Baseline | FP16 | 39.16 | 46.67 | 58.80 | 64.82 | 68.72 | 70.23 | 72.39 |
| RTN [†] | | 18.34 | 40.62 | 36.31 | 59.27 | 64.66 | 67.38 | 70.48 |
| OPTQ[†] | INT4 | **34.74** | **48.38** | 56.45 | 62.97 | 66.37 | 69.12 | 72.40 |
| **Z-FOLD** | | 33.70 | 48.03 | **57.83** | **66.83** | **69.45** | **71.14** | **72.87** |
| RTN [†] | | 0.10 | 27.36 | 0.00 | 0.00 | 0.00 | 0.06 | 1.46 |
| OPTQ[†] | INT3 | 13.93 | 32.31 | 37.26 | 52.26 | 54.98 | 64.18 | 69.69 |
| **Z-FOLD** | | **24.37** | **51.10** | **55.33** | **63.59** | **69.96** | **70.48** | **72.29** |
| RTN | | 0.00 | 0.00 | 0.00 | 0.00 | 0.00 | 0.00 | 0.00 |
| OPTQ | INT2 | 0.00 | 0.00 | 0.00 | 0.00 | 0.83 | 4.70 | 26.94 |
| **Z-FOLD** | | **6.42** | **16.67** | **28.10** | **43.10** | **51.58** | **54.69** | **65.75** |

(b) PIQA

| Method | Precision | 125M | 350M | 1.3B | 2.7B | 6.7B | 13B | 30B |
|---|---|---|---|---|---|---|---|---|
| Baseline | FP16 | 62.02 | 64.74 | 72.36 | 74.81 | 76.39 | 76.88 | 78.18 |
| RTN [†] | | 61.43 | 63.44 | 67.63 | 73.72 | 76.44 | 76.01 | 77.26 |
| OPTQ[†] | INT4 | 61.26 | 63.71 | 70.73 | 73.99 | 76.28 | 76.61 | **79.00** |
| **Z-FOLD** | | **62.35** | **64.47** | **71.33** | **75.35** | **76.39** | **76.55** | 78.07 |
| RTN [†] | | 56.09 | 60.61 | 52.77 | 51.90 | 50.49 | 52.99 | 56.37 |
| OPTQ[†] | INT3 | 59.25 | 61.32 | 68.34 | 71.38 | 73.29 | 75.24 | 77.58 |
| **Z-FOLD** | | **60.55** | **63.17** | **70.78** | **73.50** | **76.77** | **76.01** | **78.29** |
| RTN | | 50.44 | 50.76 | 49.62 | 49.56 | 49.29 | 50.38 | 47.82 |
| OPTQ | INT2 | 52.61 | 55.71 | 55.01 | 56.58 | 59.58 | 62.57 | 67.41 |
| **Z-FOLD** | | **57.62** | **56.80** | **64.15** | **64.42** | **70.78** | **69.15** | **74.92** |

(c) ARC-easy

| Method | Precision | 125M | 350M | 1.3B | 2.7B | 6.7B | 13B | 30B |
|---|---|---|---|---|---|---|---|---|
| Baseline | FP16 | 39.69 | 40.36 | 50.93 | 54.34 | 60.14 | 61.83 | 65.40 |
| RTN [†] | | 36.32 | 38.55 | 49.20 | 52.90 | 57.68 | 61.31 | 61.11 |
| OPTQ[†] | INT4 | 39.02 | 37.92 | **59.97** | 53.11 | **59.72** | 61.32 | **65.11** |
| **Z-FOLD** | | **39.02** | **38.85** | 50.17 | **53.24** | 59.64 | **61.41** | 64.81 |
| RTN [†] | | 30.43 | 36.07 | 27.97 | 26.05 | 25.04 | 30.60 | 34.22 |
| OPTQ[†] | INT3 | 36.15 | 36.91 | 46.17 | 48.19 | 53.41 | 56.82 | 59.72 |
| **Z-FOLD** | | **36.83** | **38.05** | **47.90** | **51.56** | **58.80** | **59.93** | **62.75** |
| RTN | | 25.84 | 25.72 | 25.42 | 25.34 | 25.59 | 26.05 | 25.29 |
| OPTQ | INT2 | 28.07 | 25.93 | 26.05 | 26.18 | 30.30 | 31.06 | 40.24 |
| **Z-FOLD** | | **32.41** | **31.78** | **40.74** | **40.36** | **47.31** | **49.03** | **54.80** |

#### (d) ARC-challenge

| Method | Precision | 125M | 350M | 1.3B | 2.7B | 6.7B | 13B | 30B |
|---|---|---|---|---|---|---|---|---|
| Baseline | FP16 | 22.87 | 24.06 | 29.44 | 31.31 | 34.56 | 35.75 | 38.14 |
| RTN [†] | | 22.44 | 23.81 | 24.91 | 29.18 | 32.59 | 35.24 | 35.41 |
| OPTQ[†] | INT4 | 22.95 | **24.83** | 28.24 | **30.12** | 33.70 | **34.90** | **37.80** |
| **Z-FOLD** | | **23.46** | 24.15 | **29.10** | 29.86 | **33.96** | 34.64 | 37.37 |
| RTN [†] | | 21.76 | 22.18 | 23.55 | 25.43 | 25.85 | 23.81 | 19.97 |
| OPTQ[†] | INT3 | **22.53** | **25.09** | 27.65 | 27.82 | 31.91 | 33.02 | **35.84** |
| **Z-FOLD** | | 22.35 | 23.46 | **28.07** | **28.92** | **34.04** | **33.53** | 34.98 |
| RTN | | **26.37** | **25.60** | 23.81 | 26.71 | 25.94 | 27.22 | 26.54 |
| OPTQ | INT2 | 24.40 | 24.74 | 23.98 | **26.79** | 23.81 | 24.06 | 26.54 |
| **Z-FOLD** | | 23.04 | 21.84 | **25.60** | 24.83 | **26.88** | **28.50** | **30.38** |

#### (e) StoryCloze

| Method | Precision | 125M | 350M | 1.3B | 2.7B | 6.7B | 13B | 30B |
|---|---|---|---|---|---|---|---|---|
| Baseline | FP16 | 59.96 | 63.21 | 70.78 | 71.74 | 74.60 | 76.64 | 77.28 |
| RTN [†] | | 60.02 | 63.08 | 59.13 | 70.78 | 73.65 | 74.47 | 75.37 |
| OPTQ[†] | INT4 | 59.58 | **63.46** | 69.64 | 70.46 | 73.90 | **76.19** | **77.08** |
| **Z-FOLD** | | **61.23** | 62.44 | **69.89** | **71.93** | **74.54** | 76.00 | 76.96 |
| RTN [†] | | 49.65 | 56.78 | 47.61 | 46.98 | 48.12 | 49.20 | 49.84 |
| OPTQ[†] | INT3 | 57.03 | 60.15 | 65.25 | 68.43 | 70.97 | 73.07 | 75.68 |
| **Z-FOLD** | | **59.00** | **60.92** | **68.68** | **70.66** | **74.92** | **74.79** | **76.64** |
| RTN | | 48.12 | 49.20 | 48.76 | 47.74 | 48.63 | 48.50 | 48.19 |
| OPTQ | INT2 | 51.94 | 54.23 | 53.47 | 52.51 | 54.04 | 61.17 | 65.63 |
| **Z-FOLD** | | **53.15** | **54.55** | **62.19** | **64.42** | **67.79** | **68.87** | **72.12** |

[†] Results are taken from (Frantar et al., 2023).

Table 7: Zero-shot performance of quantized BLOOM models (Accuracy ↑).

(a) LAMBADA and PIQA

| Method | Precision | LAMBADA | | | | | PIQA | | | | |
|---|---|---|---|---|---|---|---|---|---|---|---|
| | | 560M | 1.1B | 1.7B | 3B | 7.1B | 560M | 1.1B | 1.7B | 3B | 7.1B |
| Baseline | FP16 | 34.06 | 42.85 | 46.71 | 52.12 | 57.70 | 65.07 | 67.14 | 69.97 | 70.51 | 73.72 |
| RTN[†] | | 26.00 | 39.06 | 41.92 | 45.84 | 50.48 | 63.11 | 65.29 | 67.74 | 69.86 | 72.69 |
| OPTQ[†] | INT4 | 31.75 | 39.80 | 46.28 | 51.41 | 54.65 | **64.31** | 66.05 | **68.77** | 69.42 | **72.96** |
| **Z-Fold** | | **35.34** | **43.94** | **47.53** | 51.43 | **58.14** | 63.98 | **66.97** | 68.34 | **69.91** | 72.69 |
| RTN[†] | | 9.10 | 15.95 | 15.02 | 24.55 | 29.90 | 58.60 | 60.80 | 60.88 | 66.28 | 69.70 |
| OPTQ[†] | INT3 | 21.31 | 28.70 | 33.65 | 43.12 | 47.41 | 61.62 | 62.62 | 65.18 | 68.34 | **70.95** |
| **Z-Fold** | | **31.73** | **34.66** | **45.12** | **50.13** | **58.72** | **62.57** | **65.45** | **67.14** | **69.10** | 70.89 |
| RTN | | 0.00 | 0.00 | 0.00 | 0.00 | 0.00 | 51.09 | 48.53 | 49.23 | 49.73 | 50.22 |
| OPTQ | INT2 | 0.06 | 0.39 | 0.12 | 1.16 | 4.44 | 51.03 | 50.76 | 50.76 | 51.90 | 55.66 |
| **Z-Fold** | | **14.55** | **19.02** | **34.43** | **31.59** | **42.52** | **56.37** | **60.12** | **61.21** | **64.42** | **66.21** |

(b) ARC-easy and StoryCloze

| Method | Precision | ARC-easy | | | | | StoryCloze | | | | |
|---|---|---|---|---|---|---|---|---|---|---|---|
| | | 560M | 1.1B | 1.7B | 3B | 7.1B | 560M | 1.1B | 1.7B | 3B | 7.1B |
| Baseline | FP16 | 41.71 | 45.41 | 48.11 | 53.24 | 57.37 | 61.94 | 63.27 | 65.44 | 67.79 | 71.99 |
| RTN[†] | | 39.40 | 42.51 | 44.70 | 51.35 | 56.14 | 60.15 | 60.66 | 62.95 | 67.09 | 70.72 |
| OPTQ[†] | INT4 | 40.24 | **44.49** | 44.49 | **52.82** | 56.14 | 61.17 | 62.32 | 64.48 | 67.22 | **71.36** |
| **Z-Fold** | | **40.82** | 44.07 | **45.62** | 52.48 | 55.47 | **61.43** | **63.21** | **64.67** | **67.60** | 70.97 |
| RTN[†] | | **45.44** | **46.87** | 37.58 | 45.08 | 48.61 | 54.87 | 56.08 | 55.79 | 59.83 | 66.20 |
| OPTQ[†] | INT3 | 39.14 | 41.79 | 42.85 | 46.63 | 51.56 | 57.80 | 59.77 | 61.81 | 63.97 | 69.26 |
| **Z-Fold** | | 39.94 | 42.56 | **44.23** | **48.82** | **53.20** | **59.96** | **62.44** | **63.21** | **67.03** | **69.51** |
| RTN | | 26.81 | 27.10 | 27.27 | 26.56 | 27.95 | 47.29 | 46.53 | 45.83 | 46.15 | 46.40 |
| OPTQ | INT2 | 31.61 | 33.84 | 34.34 | 35.65 | 38.05 | 49.71 | 49.97 | 48.89 | 50.80 | 53.34 |
| **Z-Fold** | | **34.05** | **35.69** | **39.06** | **40.91** | **43.94** | **53.47** | **56.46** | **59.26** | **61.11** | **63.33** |

[†] Results are taken from (Frantar et al., 2023).

Table 8: Zero-shot performance of quantized LLAMA models (Accuracy ↑).

(a) ARC-challenge and HellaSwag

| Method | Precision | ARC-challenge | | | HellaSwag | | |
|---|---|---|---|---|---|---|---|
| | | 7B | 13B | 30B | 7B | 13B | 30B |
| Baseline | FP16 | 44.48 | 48.83 | 52.17 | 76.19 | 79.06 | 82.64 |
| RTN | | 43.13 | 44.48 | 52.83 | 74.17 | 77.06 | 81.54 |
| OPTQ | INT4 | 43.81 | 46.15 | 48.83 | 75.09 | 77.87 | 81.32 |
| **Z-FOLD** | | **43.81** | **47.83** | **52.84** | **75.40** | **78.25** | **82.40** |
| RTN | | 20.40 | 37.79 | 28.43 | 45.42 | 62.06 | 31.49 |
| OPTQ | INT3 | 34.78 | 34.78 | 45.82 | 68.74 | 73.15 | 78.05 |
| **Z-FOLD** | | **39.46** | **45.82** | **49.83** | **72.35** | **75.62** | **79.60** |
| RTN | | 26.42 | 28.43 | 29.10 | 26.47 | 26.10 | 25.89 |
| OPTQ | INT2 | **29.43** | 26.42 | 33.11 | 25.32 | 26.35 | 26.03 |
| **Z-FOLD** | | 28.43 | **31.44** | **38.80** | **53.87** | **57.09** | **64.53** |

(b) PIQA and OpenBookQA

| Method | Precision | PIQA | | | OpenBookQA | | |
|---|---|---|---|---|---|---|---|
| | | 7B | 13B | 30B | 7B | 13B | 30B |
| Baseline | FP16 | 79.16 | 80.14 | 82.21 | 42.60 | 43.80 | 45.60 |
| RTN | | 78.24 | 79.54 | 80.69 | 42.60 | 42.40 | 44.20 |
| OPTQ | INT4 | 78.40 | 79.76 | 81.01 | 42.40 | **44.20** | 45.80 |
| **Z-FOLD** | | **79.05** | **80.09** | **81.23** | **43.40** | 43.20 | **47.80** |
| RTN | | 65.83 | 72.20 | 65.34 | 29.40 | 35.00 | 32.00 |
| OPTQ | INT3 | 75.14 | 77.64 | 79.00 | 40.40 | 39.80 | 41.80 |
| **Z-FOLD** | | **78.02** | **79.27** | **79.05** | **40.60** | **40.80** | **45.20** |
| RTN | | 49.24 | 50.00 | 50.33 | 23.20 | 24.40 | 25.40 |
| OPTQ | INT2 | 50.49 | 50.44 | 48.97 | 24.80 | 22.80 | 23.40 |
| **Z-FOLD** | | **70.24** | **72.09** | **73.99** | **32.80** | **33.80** | **36.60** |