# OpenReview forum: "A Frustratingly Easy Post-Training Quantization Scheme for LLMs"
_EMNLP/2023/Conference — EMNLP 2023 Main_

### Official Review · Reviewer_UHNK · 2023-08-02

**Soundness:** 4

**Excitement:**

3: Ambivalent: It has merits (e.g., it reports state-of-the-art results, the idea is nice), but there are key weaknesses (e.g., it describes incremental work), and it can significantly benefit from another round of revision. However, I won't object to accepting it if my co-reviewers champion it.

**Paper Topic And Main Contributions:**

This paper proposes a post training quantization scheme by adding additional scaling factors to the input channel of the matrix. These additional scaling factors can be folded into the scaling factors on the output channel of previous layer. To learn these scaling factors that minimize the distance from the original weights, the authors use alternating least square factorization. To recover accuracy using a calibration set, the authors use technique proposed in OPTQ.

The authors evaluate their technique on BLOOM models (upto 7.1B) and OPT models (upto 30B)

**Questions For The Authors:**

1. Why are there no comparisons with LLM.int8() or SmoothQuant in Table 1 and 2?

2. What is the time it takes to learn the scalars and recover accuracy?

**Reasons To Accept:**

1. A very easy to implement idea

**Reasons To Reject:**

1. The authors dont evaluate against accepted baselines (see questions for Authors). As a result it is hard to understand the value add this work provides. The value add over OPTQ only show when we move from INT4 to INT3, which does not provide a significant improvement in terms of storage reduction.

**Reproducibility:**

4: Could mostly reproduce the results, but there may be some variation because of sample variance or minor variations in their interpretation of the protocol or method.

**Reviewer Confidence:**

3: Pretty sure, but there's a chance I missed something. Although I have a good feel for this area in general, I did not carefully check the paper's details, e.g., the math, experimental design, or novelty.

---

> ### Author Rebuttal · Authors · 2023-08-29
>
> We would like to thank the reviewer for constructive feedback.
>
> Q1. Why are there no comparisons with LLM.int8() or SmoothQuant in Table 1 and 2?
>
> A1. While LLM.int8() and SmoothQuant tried to quantize both weights and activations (i.e., W8A8), Z-Fold focuses on quantizing weights into a lower bit without the activation quantization (i.e., W4A16).
> each quantization format has its own advantages.
> Specifically, noting that the LLM inference can be divided into the context and decoding stages, the W8A8 format could be a good option for the context stage, where compute-bound GEMM operations are dominant (i.e., multiple tokens are processed jointly). Thus, W8A8 would be more suitable in the stage.
> In contrast, the W4A16 format can be better for the decoding stage.
> Since tokens are generated one-by-one in the decoding stage, memory-bound GEMV operations are dominant. Thus, reducing memory footprint by quantizing weights into a lower bit could be more beneficial than quantizing activations in this case.
> In short, it is not an apple-to-apple comparisons between two format, which were also discussed in the appendix of SmoothQuant.
> Instead, we compare our method with other approaches targeting W4A16.
>
> LLAMA-PIQA (Acc (%) $\uparrow$) |[W4A16]|7B|13B|30B
> --|--|--|--|--
> FP16||76.18|79.06|82.64
> RTN(Min-Max)||77.3*|79.1*|80.1*
> OPTQ||78.24|79.82|80.09
> LLM-QAT||78.3*|79.4*|81*
> Z-Fold+OPTQ||79.27|80.20|81.23
>
>
> LLAMA-HellaSwag (Acc (%) $\uparrow$) |[W4A16]|7B|13B|30B
> --|--|--|--|--
> FP16||76.18|79.06|82.64
> RTN(Min-Max)||72.7*|76.8*|81.6*
> OPTQ||74.25|77.90|80.85
> LLM-QAT||74*|77.7*|81.8*
> Z-Fold+OPTQ||74.61|78.28|82.13
>
>
>
> LLAMA-C4 (PPL, $\downarrow$)|[W4A16]|7B|13B|30B
> --|--|--|--|--
> FP16||7.34|6.80|6.13
> RTN(Min-Max)||8.12|7.23|6.54
> OPTQ||8.35|8.75|7.56
> LLM-QAT||7.4*|6.5*|6.5*
> Z-Fold+OPTQ||7.67|6.97|6.28
>
>
> OPT-WikiText2 (PPL, $\downarrow$) |[W4A16]|125m|350m|1.3b|2.7b|6.7b|13b|30b
> --|--|--|--|--|--|--|--|--
> WFP16||27.65|22.00|14.63|12.47|10.86|10.13|9.56
> RTN(Min-Max)||37.28|25.94|48.17|16.92|12.10|11.32|10.98
> OPTQ||31.12|24.24|15.47|12.87|11.39|10.31|9.63
> ZeroQuant-V2||36.71|25.51|19.38|17.92|11.91|10.67|10.10
> Zfold+OPTQ||30.26|22.90|14.88|12.56|10.95|10.26|9.59
>
> BLOOM-WikiText2 (PPL, $\downarrow$) |[W4A16]|560m|1.1b|1.7b|3b|7.1b
> --|--|--|--|--|--|--
> WFP16||22.42|17.69|15.39|13.48|11.37
> RTN(Min-Max)||25.90|22.00|16.97|14.76|12.10
> OPTQ||24.03|19.05|16.48|14.20|11.73
> ZeroQuant-V2||25.31|23.90|16.93|14.65|12.06
> Zfold+OPTQ||23.84|18.82|16.24|14.01|11.70
>
>
>
>
>
>
>
>
> Finally, we would like to mention that we chose OPTQ (or GPTQ) as our main counterpart and focused on adding Z-Fold to OPTQ since it is a time- and memory-efficient solution to quantize hyper-scale AI models rapidly and also achieves the SOTA performance regarding the quantization of LLMs.
> Nevertheless, we emphasize that the proposed Z-Fold can also be applied to any other quantizers such as AdaRound, BRECQ, and AdaQuant, in addition to OPTQ.
> We believe that no matter which quantizer we choose, Z-Fold could be used to enhance the quantization performance of the underlying quantizer.
> Thus, it might be more suitable to evaluate our paper by focusing on how much Z-Fold can enhance the performance of existing quantizers, not being evaluated by absolute figures.
>
> C4  (INT2), OPT models | 125M | 350M | 1.3B | 2.7B | 6.7B
> -- | -- | -- | -- | -- | --
> OPTQ | 2260.00 | 8418.00 | 4028.00 | 4388.00 | 500.70
> OPT+Zfold | 215.17 | 88.03 | 41.04 | 25.46 | 18.93
>
>
>
> WikiText2 (INT2) OPT models | 125M | 350M | 1.3B | 2.7B | 6.7B
> -- | -- | -- | -- | -- | --
> OPTQ | 4563.00 | 18687.00 | 7856.00 | 8949.00 | 2958.00
> OPT+Zfold | 255.06 | 134.18 | 46.53 | 25.30 | 16.32
>
>
>
> Q2. What is the time it takes to learn the scalars and recover accuracy?
>
> A2. When compared to the existing method (e.g., OPTQ), the proposed Z-Fold requires additional processing time to determine step size parameters $\zeta$ and $\alpha$.
> In our method, this is done by taking the alternative least squares (ALS) method (see Algorithm 1), and we set the number of iterations to 15 in our experiments.
> We measure the actual processing time to perform ALS and summarize the result as follows:
>
> Model-OPT@A100|13B|30B|66B
> --|--|--|--
> OPTQ|20.0m|44.0m|1.6h
> Zfold+OPTQ|48.7m|1.7h|3.6h
>
>
> Model-BLOOM@A100|1.7B|3B|7.1B
> --|--|--|--
> OPTQ|2.9m|5.2m|10.0m
> Zfold+OPTQ|4.7m|8.3m|16.2m

---

### Official Review · Reviewer_GYyE · 2023-08-04

**Typos Grammar Style And Presentation Improvements:** n/a
**Soundness:** 4

**Excitement:**

3: Ambivalent: It has merits (e.g., it reports state-of-the-art results, the idea is nice), but there are key weaknesses (e.g., it describes incremental work), and it can significantly benefit from another round of revision. However, I won't object to accepting it if my co-reviewers champion it.

**Missing References:**

n/a

**Paper Topic And Main Contributions:**

This paper introduces a straightforward post-training quantization scheme Z-FOLD that could enhance memory efficiency of Transformer structure, and hence enhance LLMs like OPT and BLOOM.

The main contribution of this paper is that it introduces a new approach for memory efficiency: Folding, which is used between layers to quantize weights into desirebale precision.

**Questions For The Authors:**

The results in Table 5 show that small OPT Models (125M ~ 1.3B) on LAMBADA has lower accuracy when using the Z-FOLD scheme with INT4 precision, but larger OPT Models (2.7B ~ 13B) have better results. Do you have an idea on the reasons for that?


**Reasons To Accept:**

1. The results look good, and also seem to be fairly reproducible.

2. Arguments are solid and details are included.

3. This paper also shows a certain level of novelty by proposing the Z-FOLD PTQ scheme.

**Reasons To Reject:**

I assume the operation of Z-FOLD would cause DRAM R/W, hence there should be tests to show how this method will affect the memory efficiency.



**Reproducibility:**

4: Could mostly reproduce the results, but there may be some variation because of sample variance or minor variations in their interpretation of the protocol or method.

**Reviewer Confidence:**

2: Willing to defend my evaluation, but it is fairly likely that I missed some details, didn't understand some central points, or can't be sure about the novelty of the work.

---

> ### Author Rebuttal · Authors · 2023-08-29
>
> We appreciate the reviewer's valuable comments and positive evaluation on our manuscript.
>
> Q1. Z-Fold would cause DRAM R/W.
>
> A1. While Z-Fold introduces the additional parameter $\zeta$ in the quantization, the additional cost (e.g., DRAM R/W) is not incurred by $\zeta$ at the inference time since $\zeta$ can be fused into the parameters of other layers (e.g., LayerNorm or Linear) before the inference.
> For example, when folded into the Linear layer (see Figure~5), all we need is to load fused parameters $\zeta_i\alpha_i$ instead of $\alpha_i$, which means that additional DRAM R/W operation is not incurred by $\zeta$ at the inference.
> Therefore, the total cost of Z-Fold is exactly the same as those required by other conventional per-channel quantization methods (e.g., OPTQ).
>
> Q2. Some benchmark has lower accuracy when using Z-FOLD.
>
> A2.  In our experiments, we used generic text data (i.e., excerpts from randomly crawled websites), not task-specific data, as a calibration set for the quantization to keep the zero-shot setting.
> This means that the Hessian matrix computed with the calibration set may differ from the real Hessian matrix, which leads to the inaccurate estimation of the step size. Unlike OPTQ, we use the Hessian to determine the step size as in Eq. (29). Namely, such a case may occur because the assumption does not always hold true.

---

### Official Review · Reviewer_XxAx · 2023-08-05

**Typos Grammar Style And Presentation Improvements:** 1) In equation2 6, the second $W_1^{q…
**Soundness:** 2

**Excitement:**

3: Ambivalent: It has merits (e.g., it reports state-of-the-art results, the idea is nice), but there are key weaknesses (e.g., it describes incremental work), and it can significantly benefit from another round of revision. However, I won't object to accepting it if my co-reviewers champion it.

**Missing References:**

1) GPTQ: Accurate Post-Training Quantization for Generative Pre-trained Transformers
2) QLoRA: Efficient Finetuning of Quantized LLMs
3) ZeroQuant-V2: Exploring Post-training Quantization in LLMs from Comprehensive Study to Low Rank Compensation
4) LLM-QAT: Data-Free Quantization Aware Training for Large Language Models

**Paper Topic And Main Contributions:**

This paper introduces a post-training quantization (PTQ) methodology, originally referred to as OPTQ (now known as GPTQ). Building upon GPTQ, the paper proposes a new approach for minimizing the mean squared error $||W-S \odot W_{quant}||_{F}^{2}$, where $\odot$ denotes the Hadamard product. This is achieved by decomposing the step size $S$ into two scaling factors: $\zeta$, a vector of dimensions $m \times 1$, and $\alpha$, a vector of dimensions $n \times 1$. Through iterative updates to $\zeta$ and $\alpha$, the weights $W$ can be quantized to more optimal values.

The proposed approach incorporates $\zeta$ into the weights by employing a folding technique for norm-to-linear and linear-to-linear scenarios. Additionally, this method enables fine-tuning with calibration data by leveraging GPTQ to minimize loss, thereby enhancing the overall quantization performance.

The experiments in this paper were carried out on the OPT and Bloom models, which vary in size from around 125M to 13B parameters. Employing 4-bit and 3-bit quantization, the proposed Z-Fold(+GPTQ) approach demonstrates superior performance over the RTN and GPTQ baselines across nearly all downstream tasks. These results highlight the effectiveness of the proposed methodology in achieving improved quantization outcomes for various model architectures.

**Questions For The Authors:**

1) what is the precision of variables $S$, $Z$ and $A$ in equations 8 and 10?
If $W_int$ has the same precision, why not directly determine the correct values for $W_int$ based on SGD or Adam?

2) The variable $\zeta \in R^{m \times 1}$ does not consume a lot of memory, and there are few benefits to fold $\zeta$ into previous modules, such as normalization and linear layers in Figures 4 and 5? Can you provide more specific experiments/statistics to demonstrate the claim?

3) I do not understand the feature in "Z-Fold fully utilize the feature of the Transformer structure". Does it refer to the pre-LayerNorm setting? If so, how about post-LayerNorm?

4) Please describe the hardware evaluation platform, software used, benchmarks and evaluation objectives.

5) Please provide quantified results on inference time,  GPU memory footprint etc.

6) Please provide the GPU hours required to compute $\zeta$ and $\alpha$ for various model sizes.

7) This paper's model sizes range from 125M to 30B. Most of these models can be loaded directly into an A100 GPU. Can you test your approach on a larger model, e.g., a 66B or 175B model?

8) Please extend your evaluation to cover some of the latest benchmarks like TruthfulQA and MMLU.

**Reasons To Accept:**

To minimize the objective function $||W-S \odot W_{quant}||_{F}^{2}$, the proposed approach adopts a matrix decomposition approach, representing $S$ through two vectors: $\zeta \in R^{m \times 1}$ and $\alpha \in R^{n \times 1}$. It utilizes the Min-Max and MSE methods to compute $\zeta$ and $\alpha$, respectively. This technique involves a low-rank decomposition combined with quantization, enabling efficient optimization of the weights. By breaking down the complex problem into two simpler vector representations and leveraging appropriate methods, the work provides an effective and computationally feasible approach to enhance the quantization process.

**Reasons To Reject:**

Limited novelty.
Poor evaluation. The majority of baseline results were derived from the GPTQ paper. The paper does not provide quantified results for the most recent benchmarks and models. It does not evaluate the overhead of decomposing stage and the inference performance.

**Reproducibility:**

3: Could reproduce the results with some difficulty. The settings of parameters are underspecified or subjectively determined; the training/evaluation data are not widely available.

**Reviewer Confidence:**

4: Quite sure. I tried to check the important points carefully. It's unlikely, though conceivable, that I missed something that should affect my ratings.

---

> ### Author Rebuttal · Authors · 2023-08-29
>
> We would like to thank you for the valuable feedback that will be helpful in strengthening our paper.
>
> Q1. What is the precision of variables $S$, $Z$, $A$ in equations 8 and 10? If $W_{int}$  has the same precision, why not directly determine the correct values for $W_{int}$ based on SGD or Adam?
>
> A1) Generally, Post-training quantization can be divided into two stages: 1) setting the step size and 2) Optimization, where we can optimize $W_{int}$ by gradient-based optimization using SGD or ADAM. Eq (8) and (10) describe how to set the step size with newly introduced Z (or Zeta) along with A (or alpha) that is commonly used in per-channel quantization, where $A$, $Z$, and $S=(A\circ Z)$ are FP16. After setting the step sizes, we can optimize W with a calibration set using existing algorithms such as OPTQ (or GPTQ) and AdaRound. In other words, we are contributing to the process of determining the step size, and then we can choose one of the optimization algorithms for quantization, of which we adopt OPTQ, which is a time-efficient solution. Unlike the existing algorithm, we use the Hessian to determine the step size as in Eq (29) and introduce Z (or zeta) parameters to quantize models elaborately, which summarizes our contributions.
>
>
> Q2. The variable zeta does not consume a lot of memory, and there are few benefits to fold zeta into previous modules, such as normalization and linear layers in Figures 4 and 5? Can you provide more specific experiments/statistics to demonstrate the claim?
>
> A2. While Z-Fold introduces the additional parameter $\zeta$ in the quantization, the additional cost (e.g., DRAM R/W) is not incurred by $\zeta$ at the inference time since $\zeta$ can be fused into the parameters of other layers (e.g., LayerNorm or Linear) before the inference.
> For example, when folded into the Linear layer (see Figure~5), all we need is to load fused parameters $\zeta_i\alpha_i$ instead of $\alpha_i$, which means that additional DRAM R/W operation is not incurred by $\zeta$ at the inference.
> Therefore, the total cost of Z-Fold is exactly the same as those required by other conventional per-channel quantization methods (e.g., OPTQ).
>
> Q3. I do not understand the feature in “Z-Fold fully utilizes the feature of the Transformer structure”. Does it refer to the pre-LayerNorm setting? If so, how about post-LayerNorm?
>
> A3. In our approach, we exploit the features of the Transformer structure to avoid the additional computational cost incurred by the newly introduced parameter $\zeta$.
> One such feature is the pre-LayerNorm that the reviewer pointed out.
> Specifically, we incorporated $\zeta$ of the layers right behind the normalization layer (e.g., $W_q$, $W_k$, $W_v$ and the linear layer $W_1$; see Figure 1) into the parameter $\gamma$ of the normalization layer (see Figures 3 and 4).
> The other feature that we utilized is the successive linear layers.
> Since there is no non-linear function between two linear layers (i.e., the value embedding $W_v$ and the output embedding $W_o$, and the linear layers $W_1$ and $W_2$; see Figure 1), $\zeta$ of later layers $W_o$ and $W_2$ could be folded into the previous layers $W_v$ and $W_1$, respectively (see Eqs. (26) and (28)).
> In summary, by utilizing the pre-LayerNorm architecture and the successivity of linear layers, we fused all $\zeta$'s inside the Transformer block into the previous modules offline not at inference time.
>
> When the post-LayerNorm is used or a non-linear activation function exists between linear layers (e.g., GELU in BLOOM models), it is not easy to directly fuse $\zeta$ into the previous modules.
> In such cases, we quantize unfoldable weights using only out-channel step size $\alpha$, but $\zeta$ can still be used for foldable pairs (e.g., value and output embeddings ($W_v$, $W_o$) and linear layers with ReLU activations ($W_{1}$-$\text{ReLU}$-$W_{2}$)).
>
> Finally, we would like to mention that since pre-LayerNorm Transformer structures, rather than post-LayerNorm architectures, have been prevalent in the latest AI models regardless of the domain (language or vision), the proposed Z-Fold can be also utilized for the quantization of various AI models.
>
>
> Q4. Please describe the hardware evaluation platform, software used, benchmarks and evaluation objectives.
>
> A4. All experiments have been conducted on NVIDIA A100 GPU (80~GB), and we implemented the proposed Z-Fold using PyTorch.
> We quantized pre-trained LLMs (e.g., OPT and BLOOM) with Z-Fold and used perplexity as a metric to evaluate the performance of quantized models.
> We compared our results with several public LLMs (such as OPTQ and BLOOM) and also used various zero-shot tasks (e.g., LAMBADA, PIQA, Story Cloze, and ARC) to verify the generalization performance of quantized models (see Tables 5-9).
> We will add the details of our experimental setups to the main manuscript.
>
>
> Q5. Please provide quantified results on inference time, GPU memory footprint etc.
>
> A5. Since the proposed Z-Fold does not impose any additional memory or latency (see A3), the model quantized via Z-Fold can be implemented by exploiting well-known runtime implementation (e.g., ``llama.cpp'') or various dedicated kernels without any modification.
> Furthermore, the inference time and memory footprint of Z-Fold would be exactly the same as those of other per-channel quantization schemes on which Z-Fold is applied.
> This means that if Z-Fold is used together with OPTQ, then as reported in [OPTQ paper], the obtained INT3 models would perform the inference 3.2 times faster than FP16 models.
> Also, if Z-Fold is applied to Llama, the obtained INT4 models would achieve 2.3 times reduction in the inference time on MacBook M1 (see https://github.com/ggerganov/llama.cpp).
>
>
> Q6. Please provide the GPU hours required to compute and for various model sizes.
>
> A6. When compared to the existing method (e.g., OPTQ), the proposed Z-Fold requires additional processing time to determine step size parameters $\zeta$ and $\alpha$.
> In our method, this is done by taking the alternative least squares (ALS) method (see Algorithm 1), and we set the number of iterations to 15 in our experiments.
> We measure the actual processing time to perform ALS and summarize the result as follows:
>
> Model-OPT@A100|13B|30B|66B
> --|--|--|--
> OPTQ|20.0m|44.0m|1.6h
> Zfold+OPTQ|48.7m|1.7h|3.6h
>
>
> Model-BLOOM@A100|1.7B|3B|7.1B
> --|--|--|--
> OPTQ|2.9m|5.2m|10.0m
> Zfold+OPTQ|4.7m|8.3m|16.2m
>
> Q7-Q8 Additional experiments
>
> A7-A8. We conducted more experiments as below.  We measure the quality of models when bit-width equals 2 and add the counterpart and benchmark to verify our performance as follows:
>
> llama - WikiText2 (PPL ($\downarrow$) |[W4]|7B|13B|30B|[W3]|7B|13B|30B|[W2]|7B|13B|30B
> --|--|--|--|--|--|--|--|--|--|--|--|--
> FP16||5.68|5.09|4.10||5.68|5.09|4.10||5.68|5.09|4.10
> RTN(Min-Max)||6.29|5.53|4.54||25.62|11.78|14.87||106441|57256|26805
> OPTQ||6.09|5.36|4.45||8.07|6.63|5.69||74.81|63.89|39.86
> Z-Fold+OPTQ||5.95|5.24|4.31||6.66|5.75|4.87||15.41|19.11|9.64
>
>
> llama - C4 (PPL ($\downarrow$)|[W4]|7B|13B|30B|[W3]|7B|13B|30B|[W2]|7B|13B|30B
> --|--|--|--|--|--|--|--|--|--|--|--|--
> Base||7.34|6.80|6.13||7.34|6.80|6.13||7.34|6.80|6.13
> RTN(Min-Max)||8.12|7.23|6.54||30.85|14.46|30.10||107515.00|58832.00|27869.00
> OPTQ||8.35|8.75|7.56||14.42|11.24|8.85||80.36|78.96|31.62
> Z-Fold+OPTQ||7.65|6.97|6.28||8.57|7.51|6.84||17.78|16.88|16.52
>
>
> llama - PTB (PPL ($\downarrow$)|[W4]|7B|13B|30B|[W3]|7B|13B|30B|[W2]|7B|13B|30B
> --|--|--|--|--|--|--|--|--|--|--|--|--
> Base||10.12|9.08|8.16||10.12|9.08|8.16||10.12|9.08|8.16
> RTN(Min-Max)||11.25|9.78|8.65||98.75|28.94|28.77||99310.00|80496.00|32852.00
> OPTQ||10.89|11.43|9.57||10.89|15.28|11.02||75.89|132.73|53.33
> Z-Fold+OPTQ||10.59|9.30|8.33||12.59|10.12|8.87||29.41|28.93|12.77
>
> TruthfulQA-mc1 | INT4  | 7B | 13B | 30B | INT3   | 7B | 13B | 30B
> -- | -- | -- | -- | -- | -- | -- | -- | --
> Base |  | 22.03 | 25.70 | 28.27 | | 22.03 | 25.70 | 28.27
> OPTQ | |21.79 | 24.85 | 28.52 || 20.69 | 23.75 | 23.50
> Z-Fold+OPTQ || 23.26 | 25.34 | 27.05 ||20.56 | 25.46 | 25.95
>
>
> TruthfulQA-mc2 | INT4  | 7B | 13B | 30B | INT3   | 7B | 13B | 30B
> -- | -- | -- | -- | -- | -- | -- | -- | --
> Base | |34.07 | 39.91 | 42.77 || 34.07 | 39.91 | 42.77
> OPTQ || 33.57 | 35.96 | 43.01 || 31.45 | 37.80 | 37.68
> Z-Fold+OPTQ || 35.41 | 39.90 | 42.38 || 31.18 | 39.22 | 39.19
>
>
>
> due to limited time, we were not able to complete all the experiments you suggested. We will conduct additional experiments with the latest benchmark and the results will appear in the final version.

---

### Meta-Review · Area_Chair_34ti · 2023-09-18

**Recommendation:** 4

**Metareview:**

This paper proposes a simple PTQ pipeline for the LLM quantization. Reviewers initially had concerns about missing experiments, references, and limited novelty. The authors provided a rebuttal to address them. Based on the unanimous agreement among reviewers, the AC deemed this paper should be accepted.

---

### Decision · Program_Chairs · 2023-10-07

**Decision:**

Accept-Main

**Comment:**

This paper proposes a simple PTQ pipeline for the LLM quantization. Reviewers initially had concerns about missing experiments, references, and limited novelty. The authors provided a rebuttal to address them. Based on the unanimous agreement among reviewers, the AC deemed this paper should be accepted.